# Effect of Al Content on Microstructure Evolution and Mechanical Properties of As-Cast Mg-11Gd-2Y-1Zn Alloy

**DOI:** 10.3390/ma14237145

**Published:** 2021-11-24

**Authors:** Yuanke Fu, Liping Wang, Sicong Zhao, Yicheng Feng, Lei Wang

**Affiliations:** School of Material Science and Chemical Engineering, Harbin University of Science and Technology, Harbin 150008, China; fuyuanke0209@126.com (Y.F.); lp_wang2003@126.com (L.W.); zscwr@163.com (S.Z.)

**Keywords:** magnesium alloy, microstructure, mechanical properties, LPSO phase

## Abstract

In the present paper, the Mg-11Gd-2Y-1Zn alloys with different Al addition were fabricated by the gravity permanent mold method. The effect of Al content on microstructure evolution and mechanical properties of as-cast Mg-11Gd-2Y-1Zn alloy was studied by metallographic microscope, scanning electron microscope, XRD and tensile testing. The experimental results showed that the microstructure of as-cast Mg-11Gd-2Y-1Zn alloy consisted of α-Mg phase and island-shaped Mg_3_ (RE, Zn) phase. When Al element was added, Al_2_RE phase and lamellar Mg_12_REZn (LPSO) phase were formed in the Mg-11Gd-2Y-1Zn alloy. With increasing Al content, LPSO phase and Mg_3_ (RE, Zn) phase gradually decreased, while Al_2_RE phase gradually increased. There were only α-Mg and Al_2_RE phases in the Mg-11Gd-2Y-1Zn-5Al alloy. With the increase of Al content, the grain size decreased firstly and then increased. When the Al content was 1 wt.%, the grain size of the alloy was the minimum value (28.9 μm). The ultimate tensile strength and elongation increased firstly and then decreased with increasing Al addition. And the fracture mode changed from intergranular fracture to transgranular fracture with increasing addition. When Al addition was 1 wt.%, the maximum ultimate tensile strength reached 225.6 MPa, and the elongation was 7.8%. When the content of Al element was 3 wt.%, the maximum elongation reached 10.2% and the ultimate tensile strength was 207.7 MPa.

## 1. Introduction

As the lightest metal structural material, magnesium alloys are widely used in aerospace, automobile and electronic industries for their low density, high specific strength and good casting performance and excellent machining performance [1,2,3,4]. Currently, popular magnesium alloys include Mg-Al [5], Mg-Zn [6], and Mg-RE [7] alloy. Among them, Mg-RE alloy has attracted extensive attention due to their excellent mechanical properties at room temperature and elevated temperature. As the representative of rare earth elements, Y and Gd elements have attracted the extensive attention of many scholars, and Mg-Gd-Y alloy has become the hotspot of research in recent years. However, the grain size of as-cast Mg-Gd-Y alloy is coarse, so it is necessary to take measures to refine the grain in industrial production. Recently, Zr is often used to refine the grains of Mg-RE alloys. Zr is mostly added in the form of Mg-Zr master alloy [8], which is very expensive [9]. At the same time, the density of Zr is much higher than that of magnesium, and it is easy to settle in the process of melting, which leads to a low utilization rate [10]. The grain size of Mg-Gd-Y alloy refined by Zr has poor thermal stability. The grain can be easy to grow during heat treatment [10]. All above mentioned limit the use of Zr as a refiner of Mg-RE alloy. In recent years, it has been found that Al is an effective grain refiner for Mg-RE alloy. Compared with Zr, the price of Al is much lower, and it is easier to add into Mg melt. It attracts many scholars to conduct a large number of studies on the refinement of Mg-RE alloy by Al instead of Zr. Dai [10] added Al element to Mg-10Gd alloy to generate Al_2_Gd phase, which existed stably during solution treatment and inhibited grain growth. Zhuang [11] added Al element to Mg-9Gd alloy and lamellar Al_2_Gd phase was formed in the microstructure during solution treatment. The alloy was strengthened by the different orientations of the Al_2_Gd phase within the grain and between adjacent grains. Qiu [12] found that adding Al element to Mg-10Y alloy could effectively refine α-Mg grains, and the refinement effect was the best when Al content was wt.%. Yin [13] added Al element to Mg-4Y alloy to form Al_2_Y phase, which refined the grain, and the eutectic microstructure significantly changed from α-Mg + Mg_24_Y_5_ to α-Mg + Al_2_Y. Dai [14] found that when Al was added to Mg-Gd-Y alloy, the refinement effect of Al was equal to that of Zr and as nucleating particle, Al_2_ (Gd, Y) phase could exist stably at high temperature. After adding 1.5 wt.% Al to Mg-4Y-3Nd alloy, Feng [15] found that Al_2_RE phase acting as nucleating particle could exist stably at high temperature and heat treatment could effectively control the microstructure and mechanical properties of the alloy. After 0.2–1.1 wt.% Al added to Mg-8.5Gd-5Y alloy, the LPSO phase was formed in the microstructure. With the increase of Al content, the amount of the LPSO phase gradually decreased. The ultimate tensile strength and yield strength of extruded Mg-8.5Gd-5Y-0.2Al could reach 376 MPa and 279 MPa, respectively [16]. Fang [17] added Al element to Mg-5Gd-5Y-2Zn alloy to change the content, size, and distribution of the second phase, which could effectively improve the mechanical properties of the alloy. Zhou [18] added 0.5 wt.% Al element to Mg-8Gd-4Y-1Zn could effectively improve the mechanical properties of the alloy. According to the comprehensive analysis of recent studies, it is proved that LPSO structure can appear when Zn and Al elements are simultaneously added to Mg-RE alloy [19,20,21,22]. Moreover, the LPSO phase coexists with Al_2_RE and Mg_3_ (RE, Zn) phases, which plays a synergistic strengthening and toughening effect on magnesium alloys [23,24,25]. The microstructure and strengthening effect are related to the contents of RE, Zn and Al in the alloy. At present, the studies on the addition of Al to Mg-Gd-Y alloy mainly focus on the effect and mechanism of Al refinement, while the studies on the regulation of Al on the microstructure of alloys are not enough in-depth. Therefore, with Zn and Al elements added to Mg-11Gd-2Y at the same time, the effect of different Al content on the microstructure evolution and mechanical properties of as-cast Mg-11Gd-2Y-1Zn alloy was studied.

## 2. Materials and Methods

In this experiment, a self-made Mg-11Gd-2Y-1Zn alloy was taken as the research object to study the effect of different Al content on microstructure evolution and mechanical properties (except for specific instructions, the content of chemical components involved in this paper is the mass percentage). Mg-30 wt.% Gd, Mg-30 wt.% Y, industrial pure Zn, industrial pure Mg and industrial pure Al were used as raw materials. Smelting was carried out in a resistance furnace. The steel crucible was used for melting, which was preheated by the furnace. Smelting was conducted in a protective atmosphere (0.5 Vol% SF6 + 99.5 Vol% CO_2_). When the furnace temperature rose to 780 °C, Mg was added to the crucible. Then, Mg-30 wt.% Gd and Mg-30 wt.% Y master alloys were added after all Mg was melted. When all the master alloys were melted, Zn and Al were added. After all Zn and Al were melted, the melt was stirred and homogenized for 10 min at 780 °C, and then poured into a preheated permanent mold (200 °C). The alloy ingot was removed after cooling. The actual chemical compositions of experimental alloys were determined by inductively coupled plasma optical emission spectrometer (ICP-6300, Thermo Fisher Scientific Inc., Hillsboro, OR, USA) and the test results were shown in Table 1. Metallographic specimens were prepared according to standard metallographic specimen preparation procedures, which were ground, polished and etched. The metallographic specimens for microstructure observation were etched in 4 Vol% nitric acid and ethanol for about 10 s and those for measuring grain size were etched in a picric ethanol solution (29.4 g picric acid, 41 mL water, 50 mL acetic acid and 350 mL ethanol for about 20 s. The microstructure was observed by XD30M optical microscope (Ningbo Sunny Instruments Co., Ltd., Ningbo, China) and Apreo C scanning electron microscope (Thermo Fisher Scientific Inc., Hillsboro, OR, USA). The grain size was obtained by measuring 100 grains using a linear intercept method under the OM and the phase composition was analyzed by energy dispersive spectrometer (Oxford INCA energy dispersive spectrometer (Oxford INCA energy dispersive X-ray spectrometer, Thermo Fisher Scientific Inc., Hillsboro, OR, USA). The phases were analyzed with an X-ray diffraction (X’ Pert PRO), and the test parameters were as follows: tube voltage 40 kV, tube current 40 mA, λ = 0.154056 nm, Cu Kα target, scanning angle 10°~90°, scanning speed 2θ = 8°/min. CSS-44200 universal testing machine (MTS Systems Co., Eden Prairie, MN, USA) was used to measure the mechanical properties of the specimens, and the tensile speed was 1 mm/min. The gauge size of tensile specimens was 15 mm × 5 mm × 2 mm. Five specimens of each alloy composition were taken, and the test results were averaged. The tensile fracture morphology was observed by Apreo C scanning electron microscope (Thermo Fisher Scientific Inc., Hillsboro, OR, USA).

## 3. Results and Discussion

### 3.1. Microstructure

The XRD patterns of as-cast Mg-11Gd-2Y-1Zn-xAl (x = 0, 1, 2, 3, 4, 5) are shown in Figure 1. It can be seen from Figure 1 that the microstructure consists of four phases: α-Mg phase, Mg_12_REZn phase, Mg_3_ (RE, Zn) phase and Al_2_RE phase. When Al is not added to the alloy, only the α-Mg phase and eutectic Mg_3_ (RE, Zn) phase are present in the microstructure. When the Al content is 1 wt.%, the Al_2_RE phase appears in the microstructure. With the increase of Al content, the amount of Al_2_RE phase increases while the amount of Mg_12_REZn and Mg_3_ (RE, Zn) gradually decreases. When Al content is 5 wt.%, only α-Mg and Al_2_RE phases are present in the microstructure. However, XRD can only reflect the phase components and relative content of the alloy but cannot reflect the morphology and distribution of the phases, so it is necessary to use other characterization methods to further explain the microstructure.

The SEM images of as-cast Mg-11Gd-2Y-1Zn-xAl are shown in Figure 2. It can be seen that the as-cast alloy has five phases, which are composed of black matrix phase labeled A, island-like eutectic phase labeled B, lamellar phase labeled C, irregular polygonal granular phase labeled D and acicular phase labeled E (as shown in Figure 2a,b). According to the above XRD analysis, there are four types of phases such as α-Mg, Mg_3_ (RE, Zn), Mg_12_REZn and Al_2_RE phases in the as-cast Mg-11Gd-2Y-1Zn-xAl alloy. Combined with energy spectrum analysis (as shown in Table 2), it can be determined that: A is α-Mg phase, B is Mg_3_ (RE, Zn) phase, also known as W phase, C is Mg_12_REZn phase, also known as LPSO phase, and D and E are Al_2_RE phases, which are similar to the research results of Zhou [18]. When Al is not added, the as-cast microstructure consists of α-Mg phase and Mg_3_ (RE, Zn) phase (as shown in Figure 2a). When Al content is 1 wt.%, acicular Al_2_RE phase, irregular polygonal granular Al_2_RE phase and lamellar LPSO phase appear in the microstructure (as shown in Figure 2b). With the increase of Al content, the amount of Mg_3_ (RE, Zn) phase and LPSO phase gradually decreases, while the amount of Al_2_RE phase gradually increases (as shown in Figure 2c–e). When Al content is 5 wt.%, there are only α-Mg and Al_2_RE phases (as shown in Figure 2f) in the microstructure, while Mg_3_ (RE, Zn) and LPSO phases disappear. Therefore, adjusting the Al content can significantly control the constituents and morphology of the phase.

The metallographic micrographs of as-cast Mg-11Gd-2Y-1Zn-xAl are shown in Figure 3. It can be seen that, without Al addition, the as-cast microstructure of the alloy consists of a white α-Mg matrix and black second phase distributed along grain boundary (as shown in Figure 3a), and the black second phase is all Mg_3_ (RE, Zn) phase. With 1 wt.% Al addition in Mg-11Gd-2Y-1Zn alloy, Al_2_RE phase and lamellar LPSO phase appear in the microstructure. In addition, there are two types of Al_2_RE phase. One is irregular polygonal granular, and the other is acicular. Irregular polygonal granularAl_2_RE phase appears inside the α-Mg grains. The acicular Al_2_RE phase forms at the grain boundary (as shown in Figure 3b). With the increase of Al content, the amount of Mg_3_ (RE, Zn) phase, lamellar LPSO phase and irregular polygonal granular Al_2_RE phase gradually decrease, while acicular Al_2_RE phase increases. Among them, the granular Al_2_RE phase tends to grow up (see Figure 3c–f). When Al content is 5 wt.%, the second phase in the microstructure is only Al_2_RE phase.

The effect of Al content on the grain size of as-cast Mg-11Gd-2Y-1Zn alloy is shown in Figure 4. It can be seen that, with the increase of Al content, the grain size of the alloy decreases firstly and then increases. Without Al addition, the grain size of the alloy is the largest (164.2 μm). With the increase of Al content (0–1 wt.%), the grain size of the alloy decreases sharply. The grain size of the alloy is the minimum value (28.9 μm) with 1 wt.% Al addition. With the increase of Al content (1–5 wt.%), the grain size of the alloy increases gradually. The grain size is 30.4 μm with 2 wt.% Al addition. When the Al content is more than 2 wt.%, the grain size increases obviously with the increase of Al content. The grain size reaches 48.9 μm with 4 wt.% Al addition. With the further increase of Al content, the grain size increases rapidly. When the Al content is 5 wt.%, the grain size reaches 68.8 μm, but it is still much smaller than that of without Al addition (164.2 μm).

It is generally known that there are three kinds of ternary equilibrium phases in Mg-Zn-RE alloy, such as W phase (cubic structure), I phase (icosahedral quasicrystal structure) and X phase (LPSO structure) [17,25]. Among them, I phase and X phase are believed to be effective reinforcement phases. In Mg-Zn-RE alloy, when RE/Zn is greater than 1, X phase is formed [18]. The RE/Zn of the experimental alloy in this paper is 6, but there is no LPSO phase in the microstructure (as shown in Figure 2a). This is due to two reasons. Firstly, there are two types of LPSO phase in Mg-Zn-RE alloy. Type I appears during casting, and LPSO phase in Mg-Zn-Y alloy belongs to this type. Type II does not appear in the casting process, but it appears in the subsequent heat treatment process, and LPSO phase in Mg-Zn-Gd alloy belongs to type II. LPSO phase in Mg-Gd-Zn and Mg-Gd-Y-Zn is generally considered to belong to type II [26]. Secondly, cooling rate can significantly affect the formation of LPSO phase [27]. LPSO phase formed by diffusion of elements, and LPSO phase can be formed when the ratio of Mg, RE and Zn reaches 12:1:1. In the present paper, permanent mould casting is adopted, which has a fast-cooling speed resulting in slow diffusion of elements. So, it is difficult to form LPSO phase. Without Al addition, α-Mg directly precipitates by homogeneous nucleation from the liquid phase. Only when the thermodynamic and kinetic conditions are met, α-Mg can nucleate and grow up, so the grain size of α-Mg grains is relatively large (164.2 μm, as shown in Figure 4). When Al is added to the alloy, the Gibbs free energy difference of the reaction of Al and RE, Mg and RE, Mg and Al is −14.8 KJ/mol, −7.46 KJ/mol and −4.3KJ/mol, respectively [13]. Therefore, Al and RE can react preferentially to generate granular Al_2_RE phase precipitating directly from liquid [28] (as shown in Figure 2b). The appropriately sized Al_2_RE phase can act as a nucleating particle of α-Mg phase during solidification. The α-Mg grains nucleate and grow on Al_2_RE phase (as shown in Figure 2b), which significantly refines α-Mg grains (as shown in Figure 4). This is attributed to the fact that Al_2_RE phase and α-Mg phase have similar lattice constants, and Al_2_RE phase and α-Mg phase conform to edge-edge matching (E2EM) model. Qiu [29] predicts that granular Al_2_RE is an effective refiner of α-Mg by edge-edge matching (E2EM) model, and this can be proved in subsequent experiments. With the increase of Al content (0–1 wt.%), the content of Al_2_RE phase increases, so the number of effective nucleation particles increases simultaneously [10], which enhances the grain refinement effect (as shown in Figure 4). The equilibrium partition coefficient of RE is less than 1, resulting in RE enrichment in the front of the solid–liquid interface [10,12]. This inhibits the migration of grain boundary and grain growth [12,13]. However, excessive Al addition (1–5 wt.%) consumes more RE elements, which reduces the solubility of RE in the front of solid–liquid interface, so the effect on inhibiting grain growth is weakened [14]. At the same time, excessive Al addition leads to the number density of effective nucleation particles per unit area decreases [14,15], which results in the refinement effect worse. When the Al content is 5 wt.%, RE element is exhausted (there are only α-Mg phase and Al_2_RE phase in the alloy microstructure as shown in Figure 2f), which leads to the increase of the grain size from 48.98 μm to 68.85 μm (as shown in Figure 4), so the refinement effect is deteriorated sharply. In spite of this, the grain size of 5 wt.% Al addition is still smaller than that of without Al addition (as shown in Figure 4), so Al addition in the Mg-11Gd-2Y-1Zn alloy always has a refinement effect.

In non-equilibrium solidification, solute elements tend to enrich in the front of the solid–liquid interface [18,30]. When the melt reaches eutectic temperature, Al and RE in the residual liquid phase can undergo eutectic reaction to form acicular Al_2_RE phase, which does not act as nucleating particles and distributes along the grain boundary. The Eutectic microstructure will be formed when solute elements such as Mg, Gd, Y and Zn reach the required composition of eutectic. When the ratio of Mg, Zn and RE reaches 12:1:1, LPSO phase can be formed. When the ratio of Mg, Zn and RE reaches 3:1:1, Mg_3_ (RE, Zn) phase can be formed. The content of RE element in LPSO phase is much lower than that in Mg_3_ (RE, Zn) phase, so the lower content of RE is favorable to the formation of LPSO phase. When 1 wt.% Al is added to Mg-11Gd-2Y-1Zn, the LPSO phase is formed (as shown in Figure 2b). This is because Al reacts with RE promoting the generation of Al_2_RE phase, which reduces the content of RE element, so Al can promote the formation of LPSO phase (as shown in Figure 2a,b). The solidification temperature range of LPSO phase and Mg_3_ (RE, Zn) phase is roughly the same. During non-equilibrium solidification, there is composition fluctuation in the front of solid–liquid interface, and the solute composition of the liquid in the front of solid–liquid interface cannot be determined during solidification. Therefore, it is difficult to determine the formation order of LPSO phase and Mg_3_ (RE, Zn) phase. The formation of Mg_3_ (RE, Zn) phase leads to the decrease of RE content in the surrounding liquid, which is beneficial to the formation of LPSO phase. Conversely, the formation of LPSO phase promotes the formation of Mg_3_ (RE, Zn) phase. Therefore, LPSO and Mg_3_ (RE, Zn) phases present a form of symbiotic alternate distribution (as shown in Figure 2b). With the further increase of Al content, the content of Al_2_RE phase increases gradually. In this process, the formation of Al_2_RE phase consumes a large amount of RE element, leading to the gradual decrease of the content of Mg_3_ (RE, Zn) phase and LPSO phase (as shown in Figure 2c–f). When Al content reaches 5 wt.%, RE element is exhausted, Mg_3_ (RE, Zn) phase and LPSO phase disappear, and only Al_2_RE phase exists in the microstructure (as shown in Figure 2f).

### 3.2. Mechanical Properties

Effect of different Al content on mechanical properties of as-cast Mg-11Gd-2Y-1Zn alloy is shown in Figure 5. It can be seen that the ultimate tensile strength (UTS), yield strength (YS) and elongation increase firstly and then decrease with the increase of Al content. When Al content is 0~1 wt.%, UTS and YS increase sharply with the increase of Al content. When the content of Al is 1 wt.%, the UTS and YS both reach the maximum value (225.6 MPa and 176.2 MPa, respectively), and the elongation is 7.8%. When Al content is 1~3 wt.%, UTS and YS decrease with the increase of Al content, while the elongation continues to increase. The decreasing rate of UTS and YS slow down gradually, as does the increasing rate of the elongation. When the content of Al is 3 wt.%, the elongation reaches the maximum value (10.2%), and the UTS and YS are 207.7 MPa and 127.6 MPa, respectively. When Al content is 3~5 wt.%, the variation trends of UTS, YS and elongation are similar, and the difference is that UTS, YS and elongation decrease at different rates with the increase of Al content. When Al content is 5 wt.%, the UTS, YS and elongation are 192.3 MPa, 78.6 MPa and 5.7%, respectively. Among them, the UTS and YS are not much different from those of without Al addition (181.4 MPa and 84.2 MPa, respectively), and the elongation is still higher than that of without Al addition (3.3%).

Effect of different Al content on fracture morphology and metallographic microstructure of as-cast Mg-11Gd-2Y-1Zn alloy is shown in Figure 6 and Figure 7, respectively. As can be seen that without Al addition, there are a large number of cracks and fewer dimples (as shown by arrows in Figure 6a). The fracture mode is an intergranular fracture. Cracks mainly exist in the second phases at grain boundaries (as shown by arrows in Figure 7a) and extend along grain boundaries. With the increase of Al content, the cracks on fracture surface decrease while the number of dimples increases. The distribution of dimples gradually becomes uniform. A small number of fine irregular particle phases can be seen in the dimples. Combined with the above analysis, it is determined that the particle is Al_2_RE phase (as shown in Figure 6b,c), and the fracture mode is still an intergranular fracture. Second phases at grain boundaries and within grains become initiation of cracks (as shown in Figure 7b,c). When Al content is 3 wt.%, fracture surface is covered with tiny dimples and tearing edges. The distribution of tearing edges is uniform. The spacing between adjacent tearing edges is the smallest and the depth is the deepest (as shown in Figure 6d). Elongation is the highest at this time, which is the same as the result in Figure 5. Cracks still extend along grain boundaries, especially triangular grain boundaries (as shown by arrows in Figure 6d). With the further increase of Al content, the dimples on fracture surface become shallow, and tearing edges increase. The distribution of tearing edges is not uniform. There are large cleavage planes and a large number of coarse grains on the fracture surface (as shown in Figure 6e,f), which leads to a sharp decrease in elongation and a change in fracture mode from intergranular fracture to transgranular fracture.

When Al is not added, as-cast Mg-11Gd-2Y-1Zn microstructure consists of α-Mg phase and Mg_3_ (RE, Zn) phase along grain boundaries (as shown in Figure 2a). Mg_3_ (RE, Zn) phase is a brittle eutectic phase and easy to crack under stress [25,27]. Cracks tend to initiate and propagate along grain boundaries during tensile process (as shown in Figure 6a), especially triangular grain boundaries (as shown by arrows in Figure 7a), resulting in poor mechanical properties. With the increase of Al content, as mentioned above, Al and RE elements react preferentially to form Al_2_RE phase, which significantly refines the grains (as shown in Figure 4). According to the Hall-Petch [30] formula, the smaller the grain size, the higher the strength of the alloy. When Al content is 1 wt.%, the grain size has the minimum value (28.9 μm) and there are LPSO Phases in the microstructure (as shown in Figure 2b), which can strengthen and toughen Mg-RE alloy [20,24], so the-UTS and YS reach the maximum value (225.6 MPa and 176.2 MPa, respectively).The smaller the grain size, the better the ability of coordinated deformation [13] and the higher the elongation (as shown in Figure 4 and Figure 5). So, with the increase of Al content (0~1 wt.%), the UTS, YS and elongation all increase. With the increase of Al content (1~3 wt.%), on one hand, the amount of RE consumed increases, so the effect of solid solution strengthening decreases [14]. On the other hand, the grain size increases (as shown in Figure 4), which leads to the worse refinement effect, so the UTS and YS gradually decrease [30]. With the decrease of RE content, the content of brittle Mg_3_ (RE, Zn) phase decreases (as shown in Figure 1 and Figure 2), so the elongation increases with the increase of Al content (1~3 wt.%). When Al content is 3 wt.%, the elongation reaches the maximum value (10.2%) and the fracture morphology also confirms this (as shown in Figure 6d). With the further increase of Al content, grains are coarsened sharply (as shown in Figure 4), so the UTS, YS and elongation all decrease and this is consistent with the fracture morphology (as shown in Figure 6e,f). When the content of Al is 5 wt.%, the grain size is still much smaller than that of without Al, so the elongation is much higher [13]. RE is consumed completely by 5wt.%Al addition (as shown in Figure 1 and Figure 2), so the effect of solid solution strengthening can be ignored. In the case, the contribution of fine grain strengthening and solid solution strengthening to strength just offsets each other, so the UTS and YS are not much different from those of without Al addition (as shown in Figure 5).

## 4. Conclusions

The microstructure and mechanical properties of as-cast Mg-11Gd-2Y-1Zn-xAl alloys (x = 1–5 wt.%) have been studied by material characterization methods. The addition of Al can significantly change the phase composition and content in the microstructure, and then affect the mechanical properties of the alloys. According to detailed analysis of the effect of Al content on the microstructure and mechanical properties of as-cast Mg-11Gd-2Y-1Zn-xAl alloys, the following conclusions were obtained.

The microstructure of as-cast Mg-11Gd-2Y-1Zn-xAl alloy consists of α-Mg, irregular polygonal granular Al2RE phase, island-like Mg_3_ (RE, Zn) phase, lamellar LPSO phase and acicular Al_2_RE phase. Without Al addition, there are only α-Mg phase and Mg_3_ (RE, Zn) phase, while there are only α-Mg phase and Al_2_RE phase with 5wt.% Al addition.

With the increase of Al content in as-cast Mg-11Gd-2Y-1Zn-xAl alloy, the grain size decreases firstly and then increase. When the Al content is 1 wt.%, the grain size of the alloy is the minimum value (28.9 μm).

With the increase of Al content in as-cast Mg-11Gd-2Y-1Zn-xAl alloy, the ultimate tensile strength, yield strength and elongation increase firstly and then decrease. When Al content is 1 wt.%, the ultimate tensile strength and yield strength both reach the maximum (225.6 MPa and 176.2 MPa, respectively) and the elongation is 7.8%. When Al content is 3 wt.%, the maximum elongation reaches 10.2% and the ultimate tensile strength and yield strength are 207.7 MPa and 127.6 MPa, respectively.

With the increase of Al content in as-cast Mg-11Gd-2Y-1Zn-xAl alloys, fracture mode changes from intergranular fracture to transgranular fracture.

## Figures and Tables

**Figure 1 materials-14-07145-f001:**
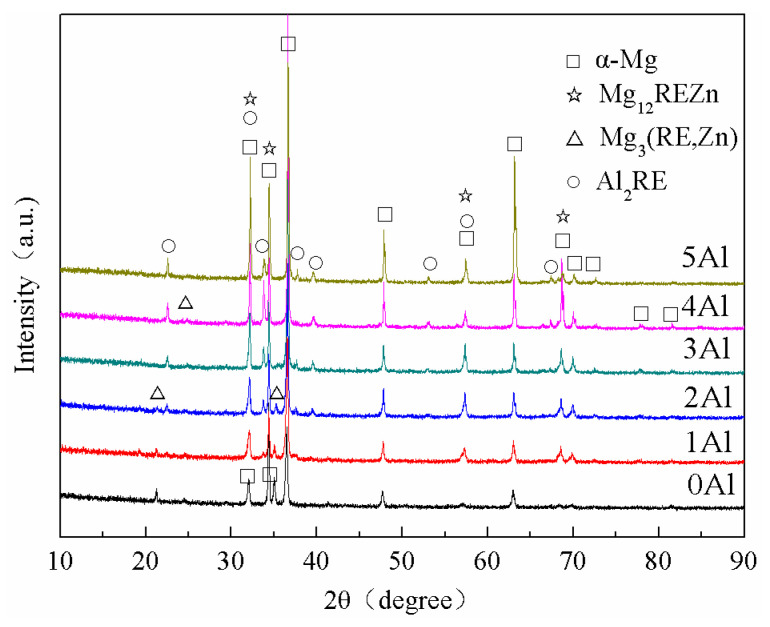
XRD patterns of as-cast Mg-11Gd-2Y-1Zn-xAl alloys.

**Figure 2 materials-14-07145-f002:**
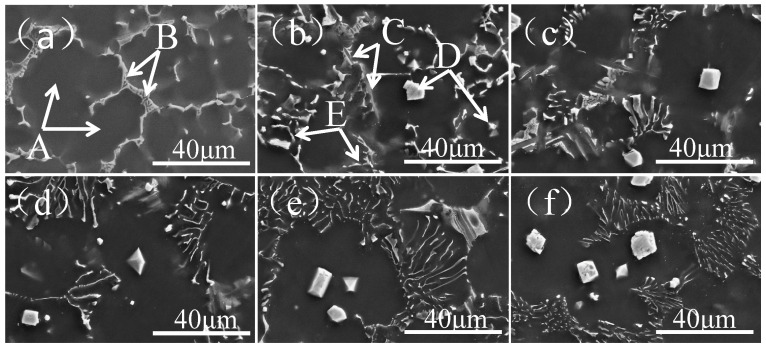
SEM images of as-cast Mg-11Gd-2Y-1Zn-xAl alloys. (**a**) x = 0; (**b**) x = 1; (**c**) x = 2; (**d**) x = 3; (**e**) x = 4; (**f**) x = 5.

**Figure 3 materials-14-07145-f003:**
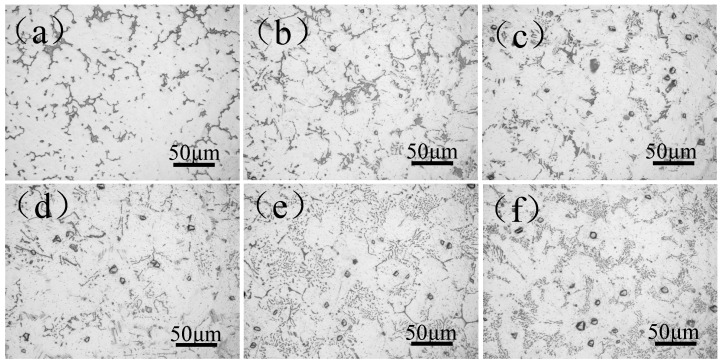
Metallographic microstructure of as-cast Mg-11Gd-2Y-1Zn-xAl alloys. (**a**) x = 0; (**b**) x = 1; (**c**) x = 2; (**d**) x = 3; (**e**) x = 4; (**f**) x = 5.

**Figure 4 materials-14-07145-f004:**
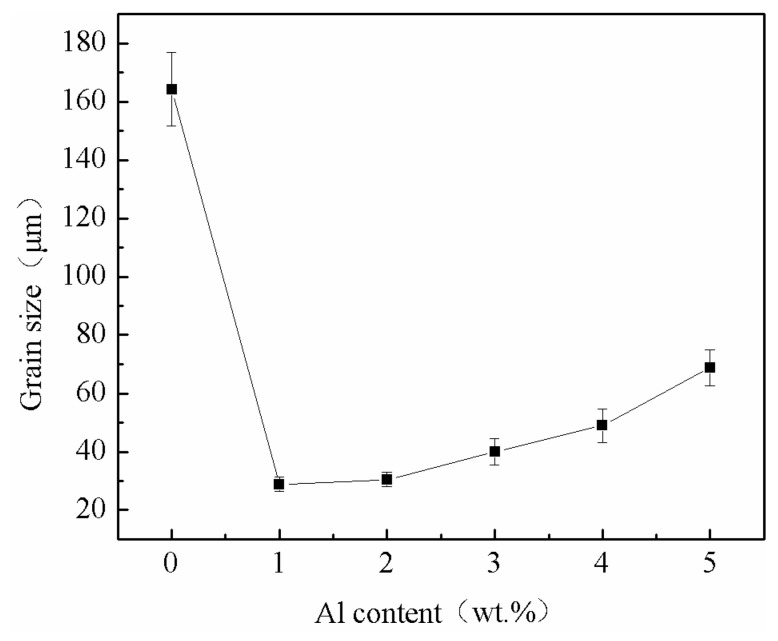
The effect of Al content on the grain size of as-cast Mg-11Gd-2Y-1Zn alloy.

**Figure 5 materials-14-07145-f005:**
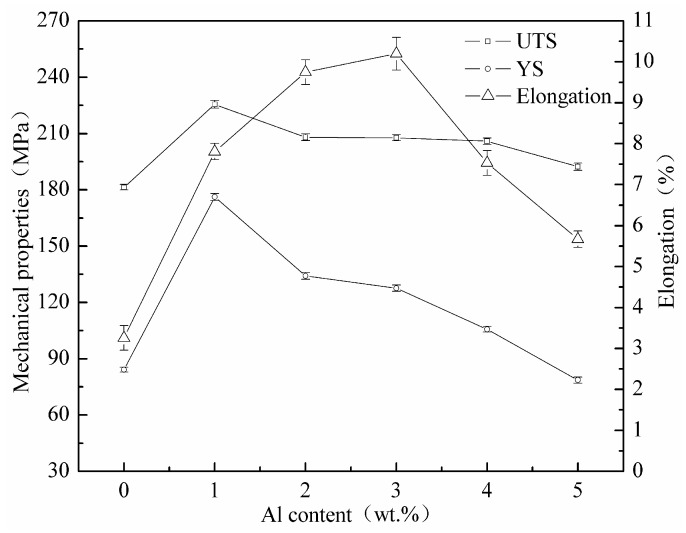
The mechanical properties of as-cast Mg-11Gd-2Y-1Zn with different Al addition.

**Figure 6 materials-14-07145-f006:**
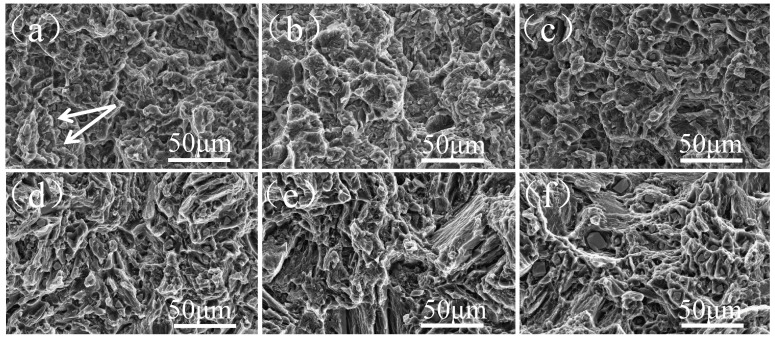
Fracture morphology of as-cast Mg-11Gd-2Y-1Zn-xAl alloys. (**a**) x = 0; (**b**) x = 1; (**c**) x = 2; (**d**) x = 3; (**e**) x = 4; (**f**) x = 5.

**Figure 7 materials-14-07145-f007:**
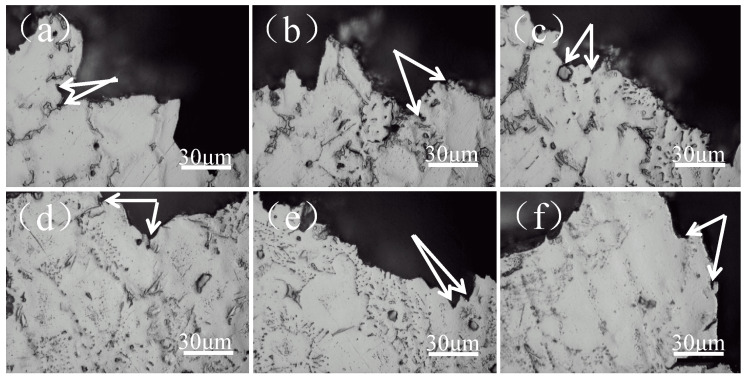
Metallographic microstructure near the fracture of as-cast Mg-11Gd-2Y-1Zn-xAl alloys. (**a**) x = 0; (**b**) x = 1; (**c**) x = 2; (**d**) x = 3; (**e**) x = 4; (**f**) x = 5.

**Table 1 materials-14-07145-t001:** Actual chemical compositions of experimental alloys (wt.%).

Alloy	Gd	Y	Zn	Al	Mg
Mg-11Gd-2Y-1Zn	10.68	2.12	0.98	0	Bal
Mg-11Gd-2Y-1Zn-1Al	11.32	1.89	1.04	1.08	Bal
Mg-11Gd-2Y-1Zn-2Al	11.26	1.86	1.02	2.06	Bal
Mg-11Gd-2Y-1Zn-3Al	10.66	2.14	1.04	2.96	Bal
Mg-11Gd-2Y-1Zn-4Al	11.16	1.88	0.96	4.05	Bal
Mg-11Gd-2Y-1Zn-5Al	11.29	2.15	0.98	4.97	Bal

**Table 2 materials-14-07145-t002:** The energy spectrum analysis results of typical second phases in Figure 2 (at.%).

Location	Mg	Gd	Y	Zn	Al	Total
B	87.09	6.72	1.35	3.65	1.18	100
C	89.61	5.03	1.20	2.50	1.65	100
D	14.9	20.16	7.65	0.53	56.76	100
E	79.85	4.45	1.72	1.16	12.83	100

## Data Availability

The data presented in this study are available on request from the corresponding author.

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
