# Peer review of "Effect of Al Content on Microstructure Evolution and Mechanical Properties of As-Cast Mg-11Gd-2Y-1Zn Alloy"

_materials, 2021, doi:10.3390/ma14237145_

Round 1

Reviewer 1 Report

1) Introduction requires additional work
This proposal is not entirely clear, additional clarifications and references are required. Comparison with iron is not correct "However, compared with iron and steel 33 materials and aluminum alloys, the mechanical properties of Mg-Gd-Y alloy are poor, 34 especially for the elevated mechanical properties."
Please Add Research Links “Recently, Zr is often used to refine 37 the grains of Mg-RE alloys. Zr is mostly added in the form of Mg-Zr master alloy, which 38 is very expensive. "
2) The method of melting described by the authors raises doubts. Usually, smelting is carried out under flux.
3) how much is the nominal composition comparable to the real one? How was the final alloy composition evaluated?
4) For a more accurate determination of the phase distribution, it would be desirable to see the EDS-maps of Figure 2. Or images at a higher magnification.
5) why the authors do not give the YS values ​​in Figure 4? This characteristic is more significant
6) the authors cite only the fact of a change in mechanical properties depending on a change in the amount of Al; however, an additional explanation of this phenomenon is required.
7) the authors do not provide data on grain size analysis
8) conclusions require adjustment

Reviewer 2 Report

Dear Authors: “Effect of Al content on microstructure evolution and mechanical properties of as-cast Mg-11Gd-2Y-1Zn alloy”. I found the article interesting. I would emphasize the utilitarian aspect in it. Unfortunately, there many shortcomings that, in my opinion, should be corrected and supplemented:

  • please highlight subchapters in the text 3.1 and 3.2,
  • In my opinion, giving point 3 as the results is not true. Because you present both the results and their analysis in it. Correct the name of this chapter.
  • Fig. 4 presents very important information about the mechanical properties of the tested alloys.
  • I did not notice in the description of the research methods how these test results were obtained. This should be completed in the article. 
  • Chapter 4 should also be marked from the text. In addition to the conclusions given, it would be good to add a few sentences of commentary that would distinguish the whole work from the research report In my opinion, the article should be improved both in terms of its structure and content 

Author Response

 “Effect of Al content on microstructure evolution and mechanical properties of as-cast Mg-11Gd-2Y-1Zn alloy”. I found the article interesting. I would emphasize the utilitarian aspect in it. Unfortunately, there many shortcomings that, in my opinion, should be corrected and supplemented:

Response: Thank you very much for your valuable suggestions. We have carefully made a revision addressing the issues in the list of response. The changes in the revised version have been highlighted.

1) Please highlight subchapters in the text 3.1 and 3.2

Response: We have highlighted subchapters in the text 3.1 and 3.2 in the revised manuscript.

2) In my opinion, giving point 3 as the results is not true. Because you present both the results and their analysis in it. Correct the name of this chapter.

Response: We have corrected the point 3 in the revised manuscript. The specific modifications are as follows:

With the increase of Al content in as-cast Mg-11Gd-2Y-1Zn-xAl alloys, fracture mode changes from intergranular fracture to transgranular fracture.

3)Fig.4 presents very important information about the mechanical properties of the tested alloys.I did not notice in the description of the research methods how these test results were obtained. This should be completed in the article.

Response: We have added the description of the research methods to the revised manuscript. The specific modifications are as follows:

CSS-44200 universal testing machine was used to measure the mechanical properties of the specimens, and the tensile speed was 1mm/min. The gauge size of tensile specimens was 15 mm×5 mm×2mm. Five specimens of each alloy composition were taken, and the test results were averaged.

4)Chapter 4 should also be marked from the text. In addition to the conclusions given, it would be good to add a few sentences of commentary that would distinguish the whole work from the research report In my opinion, the article should be improved both in terms of its structure and content 

Response: We have marked the chapter 4 in the revised manuscript and we also added the commentary to the chapter 4. In addition, after careful consideration, we have revised the topic of the chapters and the logical order of related contents. At the same time, we have added some contents to revised manuscript, such as the data and analysis of the grain size; the data and analysis of mechanical properties, especially that of yield strength; description of experimental methods and the supplement of relevant literatures.

The commentary is as follows: 

The microstructure and mechanical properties of as-cast Mg-11Gd-2Y-1Zn-xAl alloys(x=1-5wt.%) have been studied by material characterization methods. The addition of Al can significantly change the phase composition and content in the microstructure, and then affect the mechanical properties of the alloys. According to detailed analysis of the effect of Al content on the microstructure and mechanical properties of as-cast Mg-11Gd-2Y-1Zn-xAl alloys, the following conclusions were obtained.

Round 2

Reviewer 1 Report

Thanks for the detailed explanation in the revised manuscript. The article in its present form is a study suitable for publication in the journal "Materials".

Reviewer 2 Report

Thank you for making corrections and comment.